# The Influence of Steroid Therapy on the Treatment Results in Patients with Sudden Sensorineural Hearing Loss

**DOI:** 10.3390/jcm11206085

**Published:** 2022-10-15

**Authors:** Paweł Rozbicki, Jacek Usowski, Jacek Siewiera, Dariusz Jurkiewicz

**Affiliations:** 1Department of Otolaryngology and Cranio-Maxillo-Facial Surgery, Military Institute of Medicine, 04-141 Warsaw, Poland; 2Department of Hyperbaric Medicine, Military Institute of Medicine, 04-141 Warsaw, Poland

**Keywords:** sudden deafness, otorhinolaryngology, audiology, hearing loss, ssnhl, steroid therapy

## Abstract

Oral, intravenous, or intratympanic steroid therapy (ST) are commonly applied methods of pharmacotherapy in Sudden Sensorineural Hearing Loss (SSNHL). There are vast discrepancies on the recommended initial dose and the duration of ST in medical reports. The aim of the research is a retrospective comparison of patients’ examination results with different therapeutical strategies. The medical records of 218 patients treated for SSNHL at the Military Institute of Medicine were subjected to retrospective analysis by comparison of the results of pure tone audiometry prior to and after treatment with steroid therapy (first-dose delay, mass of first dose, duration of treatment). Postponement of implementation of ST to 5 days resulted in a significant improvement of hearing across all frequencies. The implementation of ST sequentially in periods (5th–10th day; >10th day) resulted in a statistically insignificant improvement of hearing in the following frequencies: 250 Hz, 1000 Hz, 1500 Hz, 2000 Hz, 3000 Hz, 4000 Hz. There was a statistical improvement of hearing within all frequencies analyzed for the initial dose of prednisone above 50 mg. For an initial dose below 50 mg, in 4000 Hz, the improvement of hearing was statistically insignificant. The research demonstrated a significant influence of steroid therapy on treatment results in patients with Sudden Sensorineural Hearing Loss.

## 1. Introduction

Sudden Sensorineural Hearing Loss (SSNHL) is defined as a sudden, sensorineural hearing loss with a raise of hearing threshold stimulus to >30 dB over at least three audiometric frequencies in Pure Tone Audiometry (PTA), developing over no longer than 72 h [1]. In pathophysiology of SSNHL, medical reports recommend autoimmune, infectious or vascular etiology. The above-mentioned factors can lead to damaged labyrinthine organs and it is measurable in cervical vestibular-evoked myogenic potentials (cVEMP) [2]. Two types of sound conduction are evaluated in PTA, namely, Bone Conduction (BC) and Air Conduction (AC). Intravenous, oral, or intratympanic steroid therapy are commonly applied methods of pharmacotherapy in patients with SSNHL. Current scientific reports do not deem any of the above-mentioned means of steroid application to be most effective in SSNHL treatment [3]. Improvement of hearing after steroid therapy in patients with SSNHL was confirmed in medical reports, including in double-blind experiment compared with placebo [1,4,5,6,7,8]. There are, however, reports questioning the effectiveness of steroid therapy in SSNHL [9]. The positive impact of steroids is attributed to their anti-inflammatory influence. Notably, steroid therapy (ST) is connected with a risk of adverse effects such as abdominal obesity, dyslipidemia, hepatic steatosis, muscle mass loss [10], a negative impact on glycemia in patients with diabetes, as well as a risk of Steroid-Induced Diabetes Mellitus (SIDM) [11]. A successful and safe steroid therapy requires a thorough broadening of knowledge on recommended treatment patterns. A study conducted by our team aims to evaluate the impact of different variants in steroid therapy on improvement of hearing in patients with SSNHL within the examined group.

## 2. Materials and Methods

Our team conducted a retrospective analysis of medical records of 218 patients (average age of 48.8 ± 14.5; 117 males and 101 females) admitted to the Clinical Department of Hyperbaric Medicine, Military Institute of Medicine due to SSNHL between January 2018 and December 2019. SSNHL was diagnosed thanks to PTA testing prior to and after the application of therapeutic patterns with the use of an Interacoustics AC40 audiometer, with the following frequencies (Hz) for given types of conducting taken into account (AC—125, 250, 500, 1000, 1500, 2000, 3000, 4000, 6000, 8000; BC—125, 250, 500, 1000, 1500, 2000, 3000, 4000). Once SSNHL was diagnosed, all test participants began steroid therapy in different schemes, and then they were referred for HyperBaric Oxygen Therapy (HBOT) with the use of a BAROXHBO (Baroxhbo, Instanbul, Turkey) device, conducted with respect to the following protocol: compression to 2.5 ATA.
-total compression/decompression time: 10 min (1.5 m/min)-oxygenation 3 × 20 min with 100% oxygen as breathing factor-oxygen breaks 2 × 5 min were performed routinely to prevent toxic influence to the lungs.

The following clinical parameters excluded participants from the tests:
-age < 18;-start of ST or HBOT 30 days after first symptoms of SSNHL occurred;-coexistent cerebrospinal inflammation;-neuropsychiatric disorder;-Meniere’s disease;-Hereditary hearing disorder;-Inner ear malformations;-Facial nerve neuroma;-Bilateral SSNHL;-Subsequent episode of SSNHL.

The differences in PTA results before and after the therapy were compared in groups with the parameters below. Due to the receiving nature of SSNHL, the BC values were primarily taken into account in calculations.


**Duration of steroid therapy**


Patients included in this stage of analysis were divided into the following groups, depending on the duration of steroid therapy: up to 16 days, longer than 16 days.


**First dose**


Patients included in this stage of analysis were divided into the following groups, depending on the first dose applied in steroid therapy, in conversion into prednisone: up to 50 mg, above 50 mg.


**First dose delay**


Patients included in this stage of analysis were divided into the following groups, depending on the interval between first symptoms of SSNHL and application of first dose in steroid therapy: up to 5 days, 5–10 days, above 10 days.

**Statistical analysis** was conducted in Microsoft Office Excel 2010 and Statistica 7.0 (Statsoft Inc., Tulsa, OK, USA). Normal distribution of the variables layout was evaluated with the visual method and a Kolmogorov–Smirnov test. The comparison within sub-groups was conducted with student’s *t*-test for independent samples. *p* < 0.05 were considered statistically significant.

## 3. Results

In statistical analysis, the examined group of patients demonstrated a layout typical for age structure and hearing depth in the Kolmogorov–Smirnov test. The characteristics of the examined group with respect to average values of hearing threshold stimulus before and after the therapy are presented in Table 1 and Table 2.

Based on the calculations above, improvement of hearing before and after the therapy was statistically significant for each of the frequencies, however, with varied power of the test. The next step of the analysis was the division of results with respect to the delay in implementing steroid therapy. For greater clarity of results, each of the variants was demonstrated in a separate column, while the differences in threshold stimulus before and after treatment were marked with the letter “d”. Student’s *t*-test for independent samples was used in the analysis.

The Table 3 demonstrates the decrease of statistical significance of improvement of hearing for sub-groups of patients with a greater delay in ST implementation. For a group with a 10-day ST implementation delay, statistically significant improvement of hearing was observed only for a frequency of 500 Hz. In the next stage, our team analyzed the improvement of hearing in a group of patients with their first dose of steroid therapy converted into prednisone at a dose not lower than 50 mg or greater than 50 mg. The results of the analysis are presented in Table 4.

The analysis of the groups above shows that in a group with a first dose in ST ≤ 50 mg for 4000 Hz, the improvement was statistically insignificant, whereas the T-value was greater in a group with ≤50 mg for audiometric frequencies 250, 500, 1500, 3000 [Hz], and lower for 1000, 2000 [Hz]. The final step of the analysis was a comparison of groups of patients with respect to the duration of steroid therapy. Sub-groups with durations of ≤16 and >16 days were distinguished. The results are presented in Table 5.

Within the examined group of patients, extending the duration of steroid therapy over 16 days was connected with lower effectiveness of treatment. A statistically significant improvement in a sub-group treated for longer than 16 days could be noticed only for frequencies 1000 and 1500 Hz, however, in both those instances the T-value was lower than in a group treated ≤16 days.

## 4. Discussion

The examined group of patients demonstrated a statistically significant improvement of hearing after treatment within the spectrum of all examined frequencies, in both types of conduction. The least statistically significant improvement was observed for a frequency of 8000 Hz in Air Conduction. This dependence seems to be in line with worse treatment results as far as the descending curves in PTA observed in current medical reports are concerned [6,12,13]. In the examined group, the statistical analysis of results confirmed statistically significant differences in the end results of treatment related to steroid therapy. The most significant improvement of hearing within the group of patients where ST was implemented was observed in medical studies of other teams [6,13,14,15]. The demonstrated dependency can be relevant for doctors in Emergency Departments, in that a quick implementation of steroid therapy can positively influence the effectiveness of treatment. A small statistical significance of hearing improvement in most of the audiometric frequencies in a group where steroid therapy was applied later than 10 days after the first symptoms occurred suggests that there is room for further scientific research aimed at determining after what period of time implementation of steroid therapy is not effective at all, meaning its application is unlikely from a forecasted improvement of hearing standpoint.

Depending on the first dose in steroid therapy, sub-groups of patients also demonstrated several differences in treatment results. In a sub-group with a first dose no greater than 50 mg converted into prednisone, an improvement proved insignificant for frequency 4000 Hz in BC. The impact of steroid doses remains controversial in current medical reports [9,16,17,18], although there are study results pointing to the benefits of implementation of high dosage steroid therapy [19,20]. An alleged effectiveness of high dosage steroids seems to be explained by a study of Niedermayer et al., which proves that patients who received an intravenous dose of 125 mg of prednisolone had a level of cortisol in perilymph similar to the control group that did not receive any steroids at all. In comparison to patients with an initial dose of 125 mg, the sub-group had a higher level of cortisol in perilymph after intravenous application of 250 mg of prednisolone [21]. Despite the above-mentioned papers, the American Academy of Otolaryngology—Head and Neck Surgery does not recommend higher doses than 60 mg of prednisone in their Clinical Practice Guideline [22].

Sub-groups of patients differing with respect to the duration of steroid therapy demonstrated significant differences in end results of the therapy. Insignificant improvement of hearing in audiometric frequencies of 250, 500, 2000, 3000, and 4000 Hz favors shorter steroid therapies in which improvement of hearing was statistically significant over all frequencies examined in Bone Conduction. The optimization of ST duration is relevant for the safety of the therapy. A prolonged steroid therapy increases the risk of adverse effects, which leads to the termination of the therapy due to those adverse effects. Scientific reports on SSNHL focus on improvement of hearing in patients exclusively; however, studies taking into account the improvement of quality of life in patients after the completion of different steroid therapies are missing, which is why further research is required.

## Figures and Tables

**Table 1 jcm-11-06085-t001:** Characteristics of hypoacusis—AC.

Frequency [Hz]	Before [dB]	After [dB]	t	Difference	*p*
125	48.99	35.33	4.045	13.66	<0.001
250	50.40	35.56	4.928	14.84	<0.001
500	54.67	36.95	5.702	17.71	<0.001
1000	54.20	37.98	5.091	16.21	<0.001
1500	67.38	39.39	5.841	27.98	<0.001
2000	54.57	40.63	4.336	13.95	<0.001
3000	59.76	46.25	3.859	13.51	<0.001
4000	60.07	48.74	3.648	11.32	<0.001
6000	66.19	54.63	3.395	11.56	<0.001
8000	63.84	57.18	2.047	6.66	<0.05

**Table 2 jcm-11-06085-t002:** Characteristics of hypoacusis—BC.

Frequency [Hz]	Before [dB]	After [dB]	t	Difference	*p*
250	41.22	26.03	4.142	15.19	<0.001
500	48.05	28.68	5.592	19.37	<0.001
1000	49.05	30.52	5.133	18.52	<0.001
1500	69.40	32.72	6.647	36.68	<0.001
2000	52.49	34.86	4.768	17.63	<0.001
3000	56.76	38.11	4.577	18.65	<0.001
4000	54.03	38.67	4.173	15.35	<0.001

**Table 3 jcm-11-06085-t003:** The delay in implementing ST and improvement of hearing in patients with SSNHL—BC.

Frequency [Hz]	<5 Days		5–10 Days		>10 Days	
	T (*p*)	d	T(*p*)	d	T (*p*)	d
250	3.41 (<0.001)	16.39	1.94 (*p* > 0.05)	14.59	1.54 (*p* > 0.05)	13.23
500	4.71 (<0.001)	21.37	2.33 (*p* < 0.05)	16.09	2.44 (*p* < 0.05)	19.88
1000	4.38 (<0.001)	20.58	2.51 (*p* < 0.05)	18.50	1.7 (*p* > 0.05)	14.29
1500	6.06 (<0.001)	43.26	2.79 (*p* < 0.01)	27.74	0.92 (*p* > 0.05)	16.50
2000	4.16 (<0.001)	20.33	2.37 (*p* < 0.05)	16.71	1.26 (*p* > 0.05)	11.25
3000	3.54 (<0.001)	18.80	2.48 (*p* < 0.05)	19.11	1.12 (*p* > 0.05)	12.42
4000	3.23 (<0.001)	15.93	2.15 (*p* < 0.05)	14.02	1.7 (*p* > 0.05)	15.18

**Table 4 jcm-11-06085-t004:** First dose in ST vs. improvement of hearing—BC.

Frequency [Hz]	≤50 mg		>50 mg	
	T (*p*)	d	T (*p*)	d
250	2.76 (<0.01)	18.57	2.59 (*p* < 0.05)	19.35
500	3.31 (<0.001)	20.74	3.04 (*p* < 0.01)	21.27
1000	2.73 (<0.01)	18.51	2.92 (*p* < 0.01)	20.40
1500	4.09 (<0.001)	41.46	3.7 (*p* < 0.001)	37.24
2000	2.62 (<0.01)	19.04	2.65 (*p* < 0.01)	18.10
3000	2.82 (<0.01)	21.44	2.49 (*p* < 0.05)	18.75
4000	1.95 (>0.05)	14.02	2.25 (*p* < 0.05)	15.06

**Table 5 jcm-11-06085-t005:** Duration of steroid therapy and improvement of hearing—BC.

Frequency [Hz]	≤16 Days		>16 Days	
	T (*p*)	d	T (*p*)	d
250	3.43 (<0.001)	19.18	1.70 (*p* > 0.05)	18.93
500	4.15 (<0.001)	21.68	1.96 (*p* > 0.05)	20.86
1000	3.39 (<0.001)	18.76	2.33 (*p* < 0.05)	23.26
1500	4.94 (<0.001)	41.34	2.54 (*p* < 0.05)	34.95
2000	3.37 (<0.001)	19.27	1.93 (*p* > 0.05)	19.04
3000	3.31 (<0.001)	20.77	1.87 (*p* > 0.05)	19.34
4000	2.43 (<0.001)	13.88	1.92 (*p* > 0.05)	17.59

## Data Availability

Data analyzed in study is present in Department of Otolaryngology and Clinical Department of Hyperbaric Medicine, Military Institute of Medicine, Warsaw.

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
