# Peer review of "The Influence of Steroid Therapy on the Treatment Results in Patients with Sudden Sensorineural Hearing Loss"

_jcm, 2022, doi:10.3390/jcm11206085_

Round 1
Reviewer 1 Report
The authors wrote an article about The influence of steroid therapy on the treatment results in patients with Sudden Sensorineural Hearing Loss. The article is very interesting, well written and the topic is very hot.
I have some suggestions to improve the quality of the article and to give a better impact in scientific literature.
1. In the introduction, please mention the correct management, the instrumental diagnosis (VEMPs-MRI) and the etiopathogenesys of the pathology, please use this reference: Ciodaro F, Freni F, Alberti G, Forelli M, Gazia F, Bruno R, Sherdell EP, Galletti B, Galletti F. Application of Cervical Vestibular-Evoked Myogenic Potentials in Adults with Moderate to Profound Sensorineural Hearing Loss: A Preliminary Study. Int Arch Otorhinolaryngol. 2020 Jan;24(1):e5-e10.
2. Please in table and result, delete the T value of t test, it is important only the p value. Re modulate the tables
3. Please in the discussion, talk about the American Academy of Otolaryngology–Head and Neck Surgery Foundation’s 2019 guidelines about SSHL.
Author Response
Dear Reviewer,
Thank you for recommendations to improve quality of manuscript. I mentioned instrumental diagnosis in introduction and I wrote about cVEMP examination. T-value is valuable for discussion because of strength of statistical test in spite of p-value - significance.
American Academy of Otolaryngology - Head and Neck Surgery is valuable paper. Addition of this paper to our discussion made our manuscript more interesting.
Best regards,

Reviewer 2 Report
Manuscript Number: jcm-1978712-peer-review-v1
Title:
The influence of steroid therapy on the treatment results in pa-2 tients with Sudden Sensorineural Hearing Loss
1. Yes, this subject is useful in Journal of Clinical Medicine
2. Author demonstrated a significant influence of steroid therapy on treatment results in patients with Sudden Sensorineural Hearing Loss.
3. The design and results are clearly presented.
4. Discussion is logical and correct.
5. Conclusion is correct.
6. References are current and pertinent. Papers of Plontke are missing.
Fill the papers of Plotnke, steroids are not recommended.
Author Response
Dear Reviewer,
Thank you for positive review of my article. I mentioned Plontke's study as a study which regard steroid therapy as a controversial.
Best regards
